# The MothersBabies Study, an Australian Prospective Cohort Study Analyzing the Microbiome in the Preconception and Perinatal Period to Determine Risk of Adverse Pregnancy, Postpartum, and Child-Related Health Outcomes: Study Protocol

**DOI:** 10.3390/ijerph20186736

**Published:** 2023-09-09

**Authors:** Naomi Strout, Lana Pasic, Chloe Hicks, Xin-Yi Chua, Niki Tashvighi, Phoebe Butler, Zhixin Liu, Fatima El-Assaad, Elaine Holmes, Daniella Susic, Katherine Samaras, Maria E. Craig, Gregory K. Davis, Amanda Henry, William L. Ledger, Emad M. El-Omar

**Affiliations:** 1UNSW Microbiome Research Centre, St George and Sutherland Clinical Campuses, UNSW Sydney, Sydney, NSW 2052, Australia; n.strout@unsw.edu.au (N.S.); l.pasic@unsw.edu.au (L.P.); chloe.hicks@unsw.edu.au (C.H.); x.chua@unsw.edu.au (X.-Y.C.); f.el-assaad@unsw.edu.au (F.E.-A.); d.susic@unsw.edu.au (D.S.); 2UNSW Stats Central, Biological Sciences South Building (E26), Level 2 Kensington, UNSW Sydney, Sydney, NSW 2052, Australia; 3Healthdirect Australia, Level 4, 477 Pitt Street, Sydney, NSW 2000, Australia; 4The Australian National Phenome Centre, Harry Perkins Institute, Murdoch University, Perth, WA 6150, Australia; elaine.holmes@murdoch.edu.au; 5Department of Women’s and Children’s Health, St George Hospital, Kogarah, NSW 2217, Australia; greg.davis@health.nsw.gov.au (G.K.D.); amanda.henry@unsw.edu.au (A.H.); 6Discipline of Women’s Health, School of Clinical Medicine, UNSW Sydney, Sydney, NSW 2052, Australia; m.craig@unsw.edu.au (M.E.C.); w.ledger@unsw.edu.au (W.L.L.); 7Complex Diseases Program, Garvan Institute of Medical Research, Darlinghurst, NSW 2010, Australia; k.samaras@garvan.org.au; 8Department of Endocrinology, St Vincent’s Hospital, Darlinghurst, NSW 2010, Australia; 9St Vincent’s Clinical Campus, UNSW Sydney, Sydney, NSW 2052, Australia

**Keywords:** microbiome, preconception, pregnancy, postpartum, perinatal health, women’s health, child health, reproduction

## Abstract

The microbiome has emerged as a key determinant of human health and reproduction, with recent evidence suggesting a dysbiotic microbiome is implicated in adverse perinatal health outcomes. The existing research has been limited by the sample collection and timing, cohort design, sample design, and lack of data on the preconception microbiome. This prospective, longitudinal cohort study will recruit 2000 Australian women, in order to fully explore the role of the microbiome in the development of adverse perinatal outcomes. Participants are enrolled for a maximum of 7 years, from 1 year preconception, through to 5 years postpartum. Assessment occurs every three months until pregnancy occurs, then during Trimester 1 (5 + 0–12 + 6 weeks gestation), Trimester 2 (20 + 0–24 + 6 weeks gestation), Trimester 3 (32 + 0–36 + 6 weeks gestation), and postpartum at 1 week, 2 months, 6 months, and then annually from 1 to 5 years. At each assessment, maternal participants self-collect oral, skin, vaginal, urine, and stool samples. Oral, skin, urine, and stool samples will be collected from children. Blood samples will be obtained from maternal participants who can access a study collection center. The measurements taken will include anthropometric, blood pressure, heart rate, and serum hormonal and metabolic parameters. Validated self-report questionnaires will be administered to assess diet, physical activity, mental health, and child developmental milestones. Medications, medical, surgical, obstetric history, the impact of COVID-19, living environments, and pregnancy and child health outcomes will be recorded. Multiomic bioinformatic and statistical analyses will assess the association between participants who developed high-risk and low-risk pregnancies, adverse postnatal conditions, and/or childhood disease, and their microbiome for the different sample types.

## 1. Introduction

The microbiome, comprising the trillions of microorganisms that live in and on us, and their collective genomes, has emerged as a key determinant of human health, including reproduction and its clinical outcomes [1]. Microbiome development starts well before birth, with emerging research demonstrating the inheritance of the microbiome across intergenerational and matrilineal lines, its association with adverse pregnancy outcomes, and its potential to impact the health of the next generation [2,3,4]. Over the last decade, the human microbiome has become established as a key contributor to human health, and it is important that the microbiome is examined not only during pregnancy, but also in the preconception period [5,6]. A person’s preconception health is a critical indicator of pregnancy outcomes, with the consequences of one’s preconception health impacting many future generations [7]. Given the unique biology surrounding conception, and the immense physiological changes during pregnancy, it would be expected that the preconception microbiome may also play a major role in successful conception, during pregnancy, and in the developing fetus [2,8,9]. The current research demonstrates there are significant links between the microbiome in pregnancy and adverse pregnancy states; however, the data are far from definitive [10,11,12,13,14]. Surprisingly, no existing perinatal microbiome research has included the critical preconception period, which has been a notable gap, and a key recommendation from expert bodies [5,6,10]. The MothersBabies Study follows on from our successful pilot study, The Microbiome Understanding in Maternity Study (MUMS) [15], which used multiomic technology to examine, longitudinally and comprehensively, the microbiomes of 100 pregnant women throughout pregnancy.

In pregnancy, many simultaneous physiological changes are required to support healthy fetal development, including hormonal, metabolic, immunomodulatory, and cardiovascular system adaptations, as well as changes in the maternal microbiome [4]. In healthy pregnant women, the microbiome changes from a balanced diverse composition in Trimester 1 to a less diverse and more pro-inflammatory microbiota in Trimester 3 [4,12]. These dramatic changes are characterized by an increased abundance of *Actinobacteria* and *Proteobacteria* phyla, the depletion of *Faecalibacterium*, and a reduction in α-diversity [4,12]. Additionally, it has been shown that there are significant differences between microbiomes in women with complicated versus uncomplicated pregnancies [12,16]. The evidence to date also suggests that, during pregnancy, the vaginal microbiome undergoes an increase in *Lactobacillus* species, a decrease in species α-diversity and β-diversity, and an increased community composition stability [17]. Notably, several research groups have shown that the vaginal microbiome profiles of women of African and European ancestry differ significantly, with microbiota-related changes occurring early in pregnancy, and they are most pronounced in women of African ancestry [18,19].

Cardiovascular and metabolic diseases, such as preeclampsia, gestational diabetes mellitus (GDM), and obesity in pregnancy, are increasing globally, and the interaction of the microbiome, and its role in their development, has become an evolving research topic [13,14,20,21,22,23]. The existing research has shown that pregnant women with preeclampsia showed a reduced bacterial diversity, with obvious dysbiosis, with the enrichment of opportunistic pathogens such as *Fusobacterium* and *Veillonella*, and the depletion of beneficial bacteria, including *Faecalibacterium* and *Akkermansia* [24]. Preliminary (unpublished) data from the MUMS Study, along with recent results from Pinto et al. have identified aberrations in the microbial composition of women with GDM that presented prior to disease onset, and persevered through gestation, indicating their potential role as non-invasive predictive biomarkers of GDM onset [25]. Additionally, the effects of the COVID-19 pandemic have seen an increase in excessive gestational weight gain (EGWG) and GDM diagnoses, with linked microbiome changes [26,27,28]. It has also been established that COVID-19 infections, in isolation, also create distinctive microbiome changes in both pregnant and non-pregnant affected individuals [26,29].

Additionally, the literature has demonstrated that there is an intergenerational and matrilineal inheritance of the birth microbiota, with microbiome development starting well before children are born [2]. The gut microbiome expands rapidly in infants following birth, eventually stabilizing at around the fifth year of age [26]. Research has already demonstrated that early dysbiosis may affect the long-term health of the gut microbiome, placing these children at an increased risk of obesity and cardiometabolic disease, allergic and atopic conditions, and adverse neurodevelopmental outcomes [27,28,29,30]. It has also been demonstrated that antibiotic usage in early life increases the risk of obesity, atopic disease, and disease later in adulthood [28,31].

Therefore, the primary objective of The MothersBabies Study is to build on these existing data, and determine whether it is possible to define preconception maternal microbiome signatures that are associated with adverse perinatal outcomes, and determine if such signatures are affected by host and environmental factors, and to what extent the preconception and perinatal microbiome is implicated in childhood health and disease.

The specific aims are:Establish the natural evolution of the maternal microbiome from preconception, through pregnancy, to the postpartum stage in multi-site microbiome communities (stool, vaginal, oral, skin, blood, and urine) in Australian women;Define the multi-site microbiome signatures (stool, vaginal, oral, skin, blood, and urine) that are associated with adverse outcomes in pregnancy (e.g., GDM, preeclampsia, perinatal mental health disorders, and obesity in pregnancy/EGWG).

The MothersBabies Study is also collecting multi-site microbiome specimens and associated metadata on all children born within the study, to correlate with their future development and health. The specific health conditions that will be examined include asthma, allergies and atopic conditions, childhood obesity, type 1 diabetes, and neurodevelopmental conditions, and we also aim to determine what is representative of a ‘healthy’ microbiome in infants, toddlers, and children.

## 2. Methods and Analysis

### 2.1. Study Design, Sample Selection, and Setting

The MothersBabies Study is a longitudinal prospective observational cohort study of 2000 mother–child pairs, commencing from preconception (up to 12 months), and continuing through to 2 years postpartum, in an Australian population. Partners can ‘opt-in’, to provide a one-off baseline sample. There are no dedicated pre-conception clinics in Australia; therefore, the cohort is self-recruited from women planning a pregnancy, using social-media advertising, as well as traditional print and television media. Participation is multi-modal—participants have the option of completing their assessments online through telehealth and questionnaires, and posting samples back to the University of New South Wales Microbiome Research Centre (UNSW MRC, Kogarah, Australia), or they can attend UNSW MRC for assessment. This ensures that the study is accessible to all participants, including those in rural and remote locations across Australia, and guarantees that the study is COVID-safe, and can continue throughout the numerous and ongoing public health orders that have been in place since the pandemic commenced in 2020.

### 2.2. Inclusion/Exclusion Criteria

The inclusion criteria are: biological female individuals, aged 18 years or older, intending to fall pregnant within 12 months of their baseline visit, intending to receive antenatal care within Australia, available for the duration of the study, and who agree to adhere to all protocol requirements, and provide written informed consent for themselves and their newborn prior to undergoing any study-related procedures. Women are excluded if they do not meet the inclusion criteria, or are pregnant at the time of enrolment. A urine human chorionic gonadotropin (HCG) test will be obtained at baseline to confirm the preconception status. In the unfortunate event that a participant experiences a pregnancy loss during the course of their participation, they have the option of leaving the study, or can continue with their 12-month preconception phase.

Participant partners are also invited to provide one-off baseline stool samples, oral samples, and vaginal samples (if applicable). There are no inclusion/exclusion criteria, and no exclusion to biological sex for this component.

### 2.3. Data Collection 

#### 2.3.1. Timeline

Data and microbiome samples will be collected across a maximum of 13 time points, as indicated in Figure 1.

#### 2.3.2. Samples for Microbiome Analysis

Microbiome samples are collected using standardised UNSW MRC clinical collection protocols that include written and video instruction. Multiple body sites (Table 1) will be analysed to create an all-encompassing microbiome ‘map’, as well as signatures for the various adverse disease outcomes in pregnancy. Microbiome samples will be self-collected by patients at selected timepoints (Table 1), and returned to the UNSW MRC for aliquoting, storage at −80 °C, and subsequent DNA extraction. Stool samples are collected using a sterile ColOff catchment bag, with the samples placed in the Stratec PSP Spin Stool DNA Plus Kit, as well as a Sarstedt tube with 95% ethanol. Oral, vaginal, and skin microbiome samples will be collected using sterile Copan FLOQswabs and eNat guanidine-thiocyanate-based DNA-stabilising medium. Samples are returned to the UNSW MRC via parcel post, in an Exempt Human Specimens B agreement with Australia Post, as the above collection devices successfully preserve microbiome samples for up to 30 days at room temperature [32,33].

#### 2.3.3. Samples for Metabolomic Analysis

Blood (serum and plasma), and first-pass urine and stool samples are collected from any participants who are able to visit UNSW MRC or a collaborating laboratory. Serum and plasma samples must be clotted for a minimum of 30 min and a maximum of 60 min prior to centrifugation at 4 °C for 15 min at 2500 rpm, before being aliquoted into microcentrifuge tubes. The urine and stool samples will be immediately aliquoted into microcentrifuge tubes. All samples will be stored at −80 °C for downstream applications.

#### 2.3.4. Clinical Data

At each visit, the participants will record their demographics; medical, surgical and obstetric history; medical conditions and any pregnancy complications that arise during the study; medications taken throughout the study; environmental exposures, including caffeine, smoking, alcohol, illicit drug use, pets and animals/livestock; sleep routine/regularity; type of conception (spontaneous or assisted and, if assisted, how); type of labour; length of labour; model of perinatal care; location of birth; and assessments collected as part of routine antenatal and postnatal care.

Children will have their type of birth, Apgar score (1, 5, and 10 min), mode of feeding, diet, sleeping environment, sleep health, physical activity and screen time, exposure to animals, illnesses, use of medications, vaccinations, and any hospital admissions recorded.

#### 2.3.5. Physical Measures

Maternal physical measures are collected during each visit, including temperature (Celsius), blood pressure (mmHg), heart rate (bpm), respiratory rate, height (cm), weight (kg), BMI, body composition, and waist circumference (cm). The child physical measures recorded at each visit post-birth include temperature (Celsius), heart rate (bpm), and respiratory rate, as well as length/height (cm), weight (kg), and head circumference (cm).

#### 2.3.6. Questionnaires

The following validated questionnaires will be used to assess maternal diet, exercise, and mood; and paediatric developmental milestones, physical activity, and sleep health.

Maternal diet is recorded using the Australian Eating Survey [34] at Baseline and Trimester 3, and a 7-day food diary (Easy Diet Diary app (https://xyris.com.au/products/easy-diet-diary/ accessed on 17 November 2021)) at all other visits. Both of these tools are validated for Australian populations.Maternal physical activity is assessed using the International Physical Activity Questionnaire [35], a publicly available, open-access tool that is useful in evaluating physical activity in large-cohort studies. The long version will be used in this study, as it provides researchers with more detailed information for comparing estimates of physical activity.Maternal mood is assessed using the Kessler Psychological Distress Scale (K10) [36], a widely recommended simple measure of psychological distress, and as a measure of outcomes following treatment for common mental health disorders, including anxiety and depression in pregnancy [37]. Due to participants completing eSurveys, this was the preferred tool over the Edinburgh Depression Scale (EDS) [38], as researchers could not guarantee participant safety should they disclose a positive response to Q10 in the EDS.The EPOCH Toddler Dietary Questionnaire (EPOCH-TDQ) [39] is a 13-item questionnaire, developed to assess aspects of dietary intake that are related to an increased risk of overweight or obesity in children aged from 1 to 2.9 years. The core questions provide data on vegetable intake, discretionary food and drink intake, and the type of grains and milk consumed. An additional set of questions (Questions 6–7) can be included, which provide data on feeding practices that may be associated with an increased obesity risk (milk intake and infant formula intake).The Ages & Stages (ASQ3) [40] developmental questionnaire, is used, and screens infant and child development in the areas of communication, gross motor, fine motor, problem-solving, and personal–social skills.The Movement Behaviour Questionnaire–Child (MBQ-C) [41] is a validated “fit-for-purpose” rapid assessment tool for measuring movement behaviours in children aged 0–5 years. The MBQ-C will be administered from the time the parent reports that the child has started walking. The MBQ-C assesses physical activity and screen time separately for weekdays and weekend days, and calculates a weighted daily average for the daily total active play, energetic play, passive screen time, and interactive screen time.Sleep health will be assessed using the MBQ-C, which assesses sleep routine in addition to sleep duration. Additionally, the Patient-Reported Outcomes Measurement Information System (PROMIS) EC Parent-Report SF v1.0-Sleep Health 8 a [42] asks eight short questions of parents to assess the quality of sleep of their children. PROMIS tools were developed in the English language, using extensive qualitative methods to ensure conceptual and semantic clarity. They have been tested for reliability and comparability against more established measures of these same content areas.

### 2.4. Impact of COVID-19 and Adaptation of Study

Not only is there emerging evidence linking microbiome changes with COVID-19, a unique period in modern history, but there were also aspects of the study that required modification [43,44,45,46,47,48,49,50]. This was to ensure participant safety during enrolment and the conduction of the study assessments. In 2020, the study received an exemption from the Australia Post service to ship biological specimens in the mail, which enabled ongoing recruitment, and continuation with existing participants, by facilitating sample collection at home, and all the study data were able to be collected digitally, using Telehealth and eSurveys. The feasibility of this had already been demonstrated, with clinical environments shifting their patient care to Telehealth for health and safety reasons. Participants’ COVID-19 infection dates, vaccination uptake, brand of vaccine, and geographical-specific lockdown data were collected. These additional data are novel, and will enable us to determine whether there are changes after COVID-19 exposure to participants, compared to pre-COVID-19 infection, and whether the microbiome recovers post-infection, whilst also establishing whether COVID-19 exposure has an impact on pregnancy outcomes through its effects on the microbiome. The MothersBabies Study’s unique design, commencing from preconception, enables researchers to monitor the effects of COVID-19 longitudinally and, for the first time, offers insight into the state of the microbiome preconception and pre-infection, providing definitive evidence of the effect that COVID-19 has during the time of infection, but also in the ensuing recovery phase.

### 2.5. Primary Outcome and Covariate Assessment

The primary outcome will be the characterisation of the maternal microbial community composition, from preconception through to 2 years postpartum, to determine whether it can successfully categorise pregnancy as either *low-risk* or *high-risk* prior to conception. A high-risk pregnancy is defined as featuring the development of any of the following conditions: preeclampsia, GDM, perinatal mental health disorders, or EGWG.

Secondary outcomes include:Examining how the pregnancy microbiome differs in women with pregnancy complications;Exploring whether the maternal microbiome is associated with excessive weight gain in pregnancy;Examining how COVID-19 infection/exposure has affected health outcomes for mothers and children;Examining how the developing paediatric microbiome in the first 5 years of life relates to the mode of birth, maternal microbiome transmission, mode of feeding, and non-communicable childhood diseases.

In addition to the primary outcomes, we will assess the following potential confounders during analysis:Pre-pregnancy BMI;Diet and physical activity;Mode and location of birth;Medication and supplements (including, but not limited to, antibiotics, prescription medication, over the counter medication, dietary supplements, pre/probiotics, vaccinations);Parity;Maternal age;Type of conception.

### 2.6. Microbiota DNA Extraction, Quantification, and Sequencing

Samples will undergo DNA extraction using the QIAGEN QIAsymphony DSP Virus/Pathogen Kit. Stool samples will be incubated at 95 °C for 10 min, and transferred to 2.0 mL bead tubes with 0.1 mm zirconia beads, before undergoing bead-beating for 5 min at 30 Hz, using TissueLyser II. Then, 25 µL Proteinase K is added to the tubes, followed by incubation for 10 min at 70 °C, then the supernatant is transferred to the QIAsymphony. Oral, vaginal, and skin swabs are incubated at 56 °C for 60 min, and undergo bead-beating and bacterial cell lysis, as above. To the oral, vaginal, and skin samples, 600 µL of buffer AL is added. DNA is quantified using the Qubit Fluorometer. Samples are stored at −80 °C until they are ready, before transfer to the UNSW Ramaciotti Centre for Genomics for shotgun metagenomic sequencing. Shotgun metagenomic sequencing has been chosen for this study as it provides greater data insight and, in addition to providing the bacteria/archaea profile, it also allows researchers to profile viruses, fungi, and other microorganisms. Moreover, it enables a higher taxonomic resolution, down to the species level, whilst providing insight into functional capabilities.

### 2.7. Metabolomic Analysis

Metabolomic analysis will be undertaken at the Australian National Phenome Centre, Murdoch University, using NMR spectroscopy and ultra-performance liquid-chromatography MS (UPLC-MS). Samples will be processed using validated sample preparation and measurement protocols for serum, urine, and faeces [51,52]. The Bruker in vitro diagnostics research (IVDr) methods, using multiple regression methods, will be used to quantify 112 lipoprotein subclasses and 34 low-molecular-weight metabolites in serum and 50 in urine [53], as well as conducting semi-quantitative untargeted analysis on a broader panel of metabolites.

### 2.8. Sample Size Calculation

As this study aims to establish whether there are microbiota differences from preconception, throughout pregnancy, and into childhood of a magnitude sufficient to justify further studies, with a view to intervention trials, the sample size is based on detecting a sizeable difference between groups in microbial β-diversity (the evolutionary distance between species). As most of the available data in and outside of pregnancy relate to the gut microbiota (versus oral or vaginal), and its possible links to cardiovascular and metabolic complications, differences in gut microbial β-diversity have been chosen for the sample size calculation. Using the effect size d (mean difference) of 0.4, based on an independent two-sample t test, to demonstrate a moderate effect size of 0.4 of gut microbiota β-diversity on pregnancy and birth outcomes, the number of disease cases required would be 51 (based on upon the pre-eclampsia, which has the lowest incidence of negative outcomes). Therefore, a total sample size of 1700 is required, with a power of 80%, and a two-sided significance level of 0.05 (alpha). Allowing for an attrition rate of 15%, comprising of failed sampling and study withdrawals, a sample size of 2000 is, therefore, sufficient to demonstrate potential clinical utility for further study.

#### Power Calculation

Power calculations for this study will be based upon the lowest incidence of negative outcomes. At present, the lowest negative outcomes are for the pre-eclampsia cohort, with a prevalence rate of 3.3% across the Australian population. *p* < 0.05 will be noted as a significant finding, and unadjusted for all initial comparisons.

### 2.9. Data Analysis Plan

The sequenced data will be processed using standard tools and pipelines that include quality control steps with Fastp [54], BBTools [55], and minimap2 [56]; the taxonomic annotation of reads to species information will be carried out via Kraken2 [57], using the GTDB (release 95, [58]) reference database, and abundances will be estimated via Bracken [59]; functional annotations will be processed via Humann3 workflow [60]. Conventional metagenomic biodiversity analyses will be performed including descriptive statistical analyses that examine the evolution of the microbiome from preconception, throughout pregnancy, to 5 years postpartum. Statistical hypothesis testing will be carried out, to compare alpha- and beta-diversities between groups of interests, such as pregnancy complications that include normal, low-risk, and high-risk pregnancy types. The detection of microbial signatures that can distinguish between groups of interests will be assessed via differential abundance analysis. Analyses will be conducted within R (R Core Team), using relevant packages, such as vegan [61] otuSummary [62], and external tools LEfSe [63]. Plots and visualisation will be generated via the R package ggplot2 [64].

### 2.10. Data Management, Ethical Procedures, and Confidentiality

Prior to the commencement of recruitment, this study obtained Human Research Ethics Committee (HREC) approval from South Eastern Sydney Local Health District (2019/ETH00192). Participants sign an eConsent form for themselves and their planned pregnancy prior to any study procedures taking place. Participants can withdraw their consent at any point in time following enrolment in the study.

The study data are collected and managed using REDCap (Research Electronic Data Capture) [65,66], hosted at UNSW. REDCap is a secure, web-based application designed to support data capture for research studies, providing an intuitive interface for validated data capture; audit trails for tracking data manipulation and export procedures; automated export procedures for seamless data downloads to common statistical packages; and procedures for data integration and interoperability with external sources [65,66].

All data collected in this study will be de-identified, and participants will be allocated a participant ID throughout their participation in this study. Only staff registered on the site delegation log, or the participant themselves via an individualised web address linked to their participant ID, will have access to this database for the inputting of information. The extraction of information will be restricted to the coordinating centre, UNSW MRC, and all information will be de-identified prior to extraction.

### 2.11. Patient and Public Involvement Statement

Consumer representatives were included in the creation of The MothersBabies Protocol, and involved in regular quarterly Steering Committee meetings. All participants in the study have access to a centralised email address, whereby feedback is regularly received and incorporated into the study design, the wording of questionnaires, and the format of visits. At the conclusion of the study, all participants will be advised of the overall results, and designated consumer representatives will be included in further discussions about the future directions of this research.

## 3. Conclusions

To date, it is only possible to test for the risk of development of adverse pregnancy outcomes, such as GDM or preeclampsia, once the woman is pregnant, or has been pregnant previously. There are no reliable pre-emptive diagnostic tools that can predict or reduce their incidence in the preconception period. The MothersBabies Study cohort will provide an invaluable, in-depth examination of the microbiome from preconception and throughout pregnancy, and over the first five years of the child’s life. This is the most comprehensive attempt ever to understand the factors that impact on pregnancy outcomes and long-term child health outcomes. The novel study format of e-visits and parcel post ensures that women across a range of environments (urban, rural, and remote communities) within Australia can participate in the study, and ensures that the results are generalizable to the entire Australian community. Moreover, the multiomic approach will enable researchers to determine the association between multi-site microbiomes and preconception risk factors. The data uncovered from this study have the potential to use the microbiome as a novel and innovative non-invasive biomarker, to characterize disease risk and personalize disease management [22,55]. This would create a substantial shift in the current clinical paradigm from symptom management to disease prevention and, whilst this might seem challenging in the short term and fraught with risk, the long-term preventative health outcomes and benefits for ongoing generations call for this research to be conducted.

## Figures and Tables

**Figure 1 ijerph-20-06736-f001:**
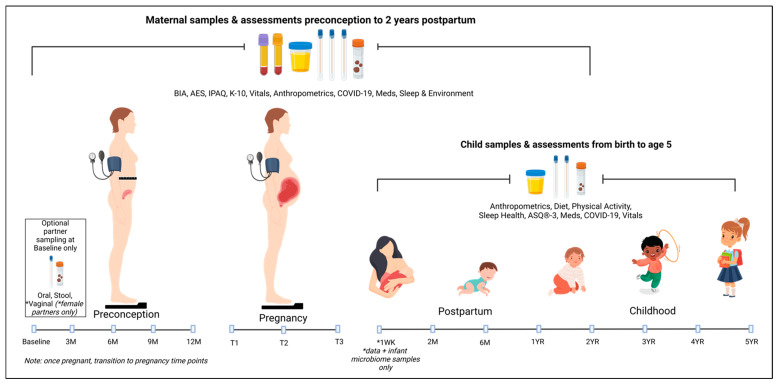
Data and microbiome samples. The maternal samples include serum and plasma, a first-pass urine sample, oral swab, vaginal swab, skin swab, and stool sample. The child samples include a first-pass urine sample, oral swab, skin swab, and stool sample. BIA: body impedance analysis. AES: Australian Eating Survey. IPAQ: International Physical Activity Questionnaire. K-10: Kessler-10. Vitals: heart rate, respiratory rate, blood pressure, temperature, and pulse oximetry. Meds: medications. ASQ: Ages & Stages Questionnaire.

**Table 1 ijerph-20-06736-t001:** Microbiome samples collected.

	Maternal	Partner	Child
Stool × 2	X	X	X
Oral	X	X	X
Skin	X		X
Urine *	X		X
Blood *	X		
Vaginal	X	X **	

* in-person visits only; ** female partners only.

## Data Availability

No new data have been created or analyzed to date, as this is a study protocol. Data sharing is not applicable to this article.

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
