# Peer review of "The MothersBabies Study, an Australian Prospective Cohort Study Analyzing the Microbiome in the Preconception and Perinatal Period to Determine Risk of Adverse Pregnancy, Postpartum, and Child-Related Health Outcomes: Study Protocol"

_ijerph, 2023, doi:10.3390/ijerph20186736_

Round 1
Reviewer 1 Report
This is an interesting topic, a well done study protocol and a new one. Hypotheses are clearly presented, as are the methods. There are only minor to moderate remarks.
1) Why don't you use a 16R-rRNA sequencing?
2) What about the costs of the study.
3) Please explain why you didn't choose developmental tests Bayley 2 and 3.
4) Please add Neumann et al, Nature Communications 2023
5) Estimated drop-out quote
Author Response
1) Why don't you use a 16R-rRNA sequencing?
16S rRNA only provides archaea and bacteria profiles and does not provide viral or fungal profiles. It can only resolve taxonomy to genus level with and no functional capablity. Shotgun metagenomic sequencing has been chosen for this study as it provides greater data insight, and in addition to bacteria/archaea profile it allows researchers to also profile virus, fungi and other microorganisms. It can resolve to higher taxonomic resolution down to species and sometimes strain level. Moreover, shotgun metagenomic sequencing also provides insight into the functional capabilities.
2) What about the costs of the study.
This study is supported by grants from the Australian Government Medical Research Future Fund Accelerated Research scheme (Grant Award: GA33765), and from St George & Sutherland Medical Research Foundation.
3) Please explain why you didn't choose developmental tests Bayley 2 and 3.
The Ages and Stages Questionnaire is a brief instrument that takes approximately 5-10 minutes for the parent to complete. The Bayley Scales of Infant and Toddler Development however are clinician-completed assessments, which require evaluation in a research office clinician room, with each assessment lasting between 30-70minutes to complete. It was therefore determined that the Ages and Stages Questionnaire would impact the least on time restraints of families, but also allowed all parents of children born on this study to complete a child-development assessment tool, without requiring attendance at a paediatrician office. This increases equitability and accessibility to research over the vast geographical expanse of Australia.
Additionally, our pilot study, The Microbiome Understanding in Maternity Study (MUMS), uses the Ages and Stages Questionnaire. For consistency across both studies, it was decided the ASQ be used for this study also.
4) Please add Neumann et al, Nature Communications 2023
We could not quite locate this reference, but would be happy to consider it if the reviewer would kindly share it with us.
5) Estimated drop-out quote
The estimated attrition rate was 15%, and this has been included in the power calculation section (line 339).
Reviewer 2 Report
The manuscript, entitled “The MothersBabies Study, an Australian prospective cohort study analyzing the microbiome in the preconception and perinatal period to determine risk of adverse pregnancy, postpartum & child-related health outcomes: study protocol” described that 2000 Australian women were recruit to explore the role of the microbiome in the development of adverse perinatal outcomes. This study is meaning, however, due to the results section is not available, my suggestion is rejection.
Comments:
1. After read the abstract, introduction, and materials and methods sections, I believe this study is meaningful and interesting. However, the data of results were not shown in this manuscript, so, this manuscript cannot be published in the journal. I suggest the authors to submit the manuscript when the study finished. Although the results section is not mandatory for the paper the type of which belongs to study protocol, strictly speaking this manuscript does not belong to study protocol, but rather to a retrospective study. It is difficult to judge the reliability of the protocol without results.
2. According to the methods shown in this study, the authors are going to follow a sample for almost 7 years, and the authors plan to collect 2000 samples. The workload is considerable, which is difficult to finish. In section of result, it is stated that final data collection and analysis of results is anticipated for 2025. So, have the authors finished most of work already? If the authors could provide some basic information of 2000 women?
3. The authors also try to study the impact of COVID-19, but how could the author prove the change of microbiome in the body caused by COVID-19?
4. What is the geographical distribution of these 2000 women in Australia? The authors should clarify this information.
Moderate editing of English language required
Author Response
The manuscript, entitled “The MothersBabies Study, an Australian prospective cohort study analyzing the microbiome in the preconception and perinatal period to determine risk of adverse pregnancy, postpartum & child-related health outcomes: study protocol” described that 2000 Australian women were recruit to explore the role of the microbiome in the development of adverse perinatal outcomes. This study is meaning, however, due to the results section is not available, my suggestion is rejection.
Response:
Our manuscript comes under the category of Study Protocols. The Guest Editors' Comments state that the authors were advised "You do not need to include a result section unless you have some preliminary or baseline data you wish to describe.”
As this is a study protocol, which can only be published before results are available, and we do not have preliminary or baseline data, it is not possible to include results in this manuscript.
Replies to comments:
After read the abstract, introduction, and materials and methods sections, I believe this study is meaningful and interesting. However, the data of results were not shown in this manuscript, so, this manuscript cannot be published in the journal. I suggest the authors to submit the manuscript when the study finished.
Please see comments above.
Although the results section is not mandatory for the paper the type of which belongs to study protocol, strictly speaking this manuscript does not belong to study protocol, but rather to a retrospective study. It is difficult to judge the reliability of the protocol without results.
This is a prospective cohort study that examines the microbiome of women as they progress from preconception to pregnancy, and then for 5 years postpartum.
According to the methods shown in this study, the authors are going to follow a sample for almost 7 years, and the authors plan to collect 2000 samples. The workload is considerable, which is difficult to finish. In section of result, it is stated that final data collection and analysis of results is anticipated for 2025. So, have the authors finished most of work already? If the authors could provide some basic information of 2000 women?
This study is ongoing, and those results are not yet available as the study is yet to finish recruitment. Publishing of study results prior to the study protocol being accepted for publication nullifies the manuscript as a study protocol. This was an anticipated date, that is dynamic and may change dependent on recruitment success.
The authors also try to study the impact of COVID-19, but how could the author prove the change of microbiome in the body caused by COVID-19?
There is emerging evidence linking microbiome changes with COVID-19. The MothersBabies Study unique design, commencing from preconception, enables researchers to monitor the effects of COVID-19 longitudinally, and for the first time, offer insight into the state of the microbiome preconception and pre-infection, providing definitive evidence of the effect COVID-19 has at the immediate time of infection, but also in the ensuing recovery phase. The effects of the COVID-19 pandemic have seen an increase in excessive gestational weight gain and GDM diagnoses, with linked microbiome changes, as demonstrated by the following literature:
- Zanardo V, Tortora D, Sandri A, Severino L, Mesirca P, Straface G. COVID-19 pandemic: Impact on gestational diabetes mellitus prevalence. Diabetes Res Clin Pract. 2022;183:109149.
- Khan MA, Moverley Smith JE. "Covibesity," a new pandemic. Obes Med. 2020;19:100282.
- Butera, A.; Maiorani, C.; Morandini, A.; Simonini, M.; Colnaghi, A.; Morittu, S.; Barbieri, S.; Ricci, M.; Guerrisi, G.; Piloni, D.; et al. Assessment of Oral Microbiome Changes in Healthy and COVID-19-Affected Pregnant Women: A Narrative Review. Microorganisms 2021, 9, 2385. https://doi.org/10.3390/microorganisms9112385
- Finlay, B.B.; Amato, K.R.; Azad, M.; Blaser, M.J.; Bosch, T.C.G.; Chu, H.; Dominguez-Bello, M.G.; Ehrlich, S.D.; Elinav, E.; Geva-Zatorsky, N.; et al. The hygiene hypothesis, the COVID pandemic, and consequences for the human microbiome. Proceedings of the National Academy of Sciences 2021, 118, e2010217118, doi:10.1073/pnas.2010217118.
- Yun Kit, Y.; Tao, Z.; Grace Chung-Yan, L.; Fen, Z.; Qin, L.; Amy, Y.L.L.; Arthur, C.K.C.; Chun Pan, C.; Eugene, Y.K.T.; Kitty, S.C.F.; et al. Gut microbiota composition reflects disease severity and dysfunctional immune responses in patients with COVID-19. Gut 2021, 70, 698, doi:10.1136/gutjnl-2020-323020.
- Etti, M.; Alger, J.; Salas, S.P.; Saggers, R.; Ramdin, T.; Endler, M.; Gemzell-Danielsson, K.; Alfvén, T.; Ahmed, Y.; Callejas, A.; et al. Global research priorities for COVID-19 in maternal, reproductive and child health: Results of an international survey. PLOS ONE 2021, 16, e0257516, doi:10.1371/journal.pone.0257516.
- Romano-Keeler, J.; Zhang, J.; Sun, J. COVID-19 and the neonatal microbiome: will the pandemic cost infants their microbes? Gut Microbes 2021, 13, 1-7, doi:10.1080/19490976.2021.1912562.
- Du, M.; Yang, J.; Han, N.; Liu, M.; Liu, J. Association between the COVID-19 pandemic and the risk for adverse pregnancy outcomes: a cohort study. BMJ Open 2021, 11, e047900, doi:10.1136/bmjopen-2020-047900.
What is the geographical distribution of these 2000 women in Australia? The authors should clarify this information.
Recruitment is open to all women, regardless of their geographical location in Australia, as the study utilises Telehealth services, eVisits and the Australian Postal system. This study is ongoing, and as such, final geographical distribution are not yet available.
Round 2
Reviewer 2 Report
The authors have revised the text, however, due to the comments unaddressed, my suggestion is rejection.
Comments:
1. The aim of this study is to establish the natural evolution of the maternal microbiome from preconception, pregnancy to postpartum in multi-site microbiome communities in Australian women. And to define the multi-site microbiome signatures that are associated with adverse outcomes of pregnancy. However, there is very little mention in the paper regarding how the collected microbial samples will be analyzed, identified, and studied. In my opinion, this aspect is crucial for the overall research.
2. Having read the abstract, introduction, and materials and methods sections, I find this study to be meaningful and intriguing. However, the absence of result data in this manuscript renders it unsuitable for publication in the journal. I recommend that the authors submit the manuscript once the study is complete. Although the results section may not be obligatory for a paper categorized as a study protocol, strictly speaking, this manuscript falls more under the retrospective study category rather than a study protocol. Without the inclusion of results, it becomes challenging to assess the reliability of the protocol.
Author Response
Comment 1: The aim of this study is to establish the natural evolution of the maternal microbiome from preconception, pregnancy to postpartum in multi-site microbiome communities in Australian women. And to define the multi-site microbiome signatures that are associated with adverse outcomes of pregnancy. However, there is very little mention in the paper regarding how the collected microbial samples will be analysed, identified, and studied. In my opinion, this aspect is crucial for the overall research.
We thank the reviewer for his/her query. We wish to point out the specific mention in the manuscript regarding the requested information: Lines 86 – 100 outline existing microbiome research linked to adverse pregnancy outcomes. The nature of samples for microbiome analysis, the timepoints collected, and the collection devices are included on lines 167 – 180, and in Table 1 and Figure 1. Lines 183 – 191 outline metabolomic processes. Lines 301 – 337 detail microbiota DNA extraction, quantification, and sequencing, and metabolomic analysis. Lines 356 – 370 specifically discuss that ‘sequenced data will be processed using standard tools and pipelines that include quality control steps with Fastp [54], BBTools [55] and minimap2 [56]; taxonomic annotation of reads to species information will be carried out by Kraken2 [57] using the GTDB (release 95, [58]) reference database and abundances are estimated by Bracken [59]; and functional annotations will be processed by Humann3 workflow [60]. Conventional metagenomic biodiversity analyses will be performed including descriptive statistical analyses that examine the evolution of microbiome from preconception throughout pregnancy to 5-years postpartum.’
We trust that this provides sufficient details to address the reviewer’s request.
Comment 2: Having read the abstract, introduction, and materials and methods sections, I find this study to be meaningful and intriguing. However, the absence of result data in this manuscript renders it unsuitable for publication in the journal. I recommend that the authors submit the manuscript once the study is complete. Although the results section may not be obligatory for a paper categorized as a study protocol, strictly speaking, this manuscript falls more under the retrospective study category rather than a study protocol. Without the inclusion of results, it becomes challenging to assess the reliability of the protocol.
We thank the reviewer for the comment. However, as per suggestions from IJERPH Editorial Office, please note the following for submitting study protocols:
‘Study Protocol: Study Protocol manuscripts report proposed or ongoing prospective research. The publication of study protocols can reduce publication bias and improve the reproducibility of research. This also helps to prevent the unnecessary duplication of effort and work. Preference will be given to submissions describing long-term studies and those likely to generate a considerable amount of outcome data. If data collection is complete, the manuscript will not be considered. Study protocols for pilot or feasibility studies, or if the authors have other articles relating to the protocol published or under consideration, are also not considered.’
MDPI does not contain a description of study protocols in it’s submission types, however BMJ, BMC and PLoS all state that:
‘Study protocols for proposed or ongoing prospective clinical research should provide a detailed account of the hypothesis, rationale and methodology of the study, and the associated ethical requirements. Protocol manuscripts should report planned or ongoing research studies, that have not yet generated results. If data collection is complete, many journals will not consider the manuscript.’
Prospective studies are defined as studies that watch for outcomes, such as the development of a disease, during the study period and relates this to other factors. This allows participants to be followed over time, with the data collected as their characteristics or circumstances change. Retrospective studies however seek to assess individuals using information collected about their past, using recall, registries, or databases. As we stated in our previous response to comments, this study is a prospective cohort study that examines the microbiome of women as they progress from preconception to pregnancy, and then for 5 years postpartum. This therefore makes this study a prospective, not retrospective study.
As this prospective study continues, and this is a Study Protocol submission, the inclusion of results is not applicable to the submission.